# Changing patterns and biological features of community-acquired *Clostridioides difficile* infection in Southwest China: 7 years of surveillance data

Wenpeng Gu,[1,2] Feng Liao,[3] Lulu Bai,[4,5] Wenzhu Zhang,[4,5] Senquan Jia,[1] Junrong Liang,[4] Yongming Zhou,[1] Jianwen Yin,[1] Xiaoqing Fu,[1] Yuan Wu[4,5]

**ABSTRACT**  The molecular epidemiological features of community-acquired *Clostridioides difficile* infection in Southwest China from 7 years of surveillance data were analyzed. Four representative *C. difficile* strains were selected for RNA-seq, biofilm formation, toxin expression, and cytotoxicity assays. Overall, 5.04% of the *C. difficile* strains were isolated within 7 years, 85.51% of which were toxigenic *C. difficile* (both *tcdA+/tcdB+*). Multilocus sequence typing (ST) and genomic sequencing divided all the isolates into two clusters, namely, clade 1 and clade 4, respectively. ST37 of *C. difficile* gradually replaced the ST3, ST35, and ST54 genotypes and became the dominant genotype in this area. The antibiotic resistance rate of strains in clade 4 was higher than that in clade 1, especially for the ST37 genotype strains, which were resistant to quinolones. Four *C. difficile* strains, R20291 (RT027), CD21062 (RT078), CD279 (ST54), and CD413 (ST37), were selected as representative isolates for subsequent biological investigations. RNA-seq revealed that the DEGs of *C. difficile* ST54 were enriched mainly in ABC transporters, two-component systems, and quorum sensing (QS) pathways and exhibited strong biofilm formation ability. The DEGs of the ST37 genotype strains were mainly enriched in the phosphotransferase system (PTS), ribosome, and some sugar and amino acid metabolism pathways, suggesting that these isolates have increased proliferation and metabolic status. On the other hand, *C. difficile* R20291 had the highest level of toxin transcription, expression, and cytotoxicity among these four strains. These genotype strains had their own biological characteristics, which provided certain clues for analyzing the causes of these changes.

**IMPORTANCE**  This study carried out a molecular epidemiological investigation of community-acquired *C. difficile* infection in Southwest China and revealed the characteristics of genotype pattern changes in the strains. *C. difficile* ST37 gradually replaced the ST3, ST35, and ST54 genotypes to become the dominant strains in this area. Moreover, some representative strains were used to study their biological features. The ST54 strain had strong biofilm formation ability, and ABC transporters, two-component systems, and quorum sensing pathways were enriched according to RNA-seq. The ST37 genotype strain was enriched in the PTS, ribosome, and several sugar and amino acid metabolism pathways. The antibiotic resistance rate of Clade 4 *C. difficile* was higher than that of clade 1 strains, especially for the resistance of *C. difficile* ST37 to quinolones. The biological characteristics of these representative strains might provide certain clues for investigating the reasons for these changes.

**KEYWORDS**  *Clostridioides difficile*, pattern changes, RNA-seq, biological features

**Peer Reviewers** Maria Radoslavova Pavlova, National Center of Infectious and Parasitic Diseases, Sofia, Bulgaria; Jianxin Wang, Beijing Key Laboratory of Emerging Infectious Diseases, Institute of Infectious Diseases, Beijing Ditan Hospital, Capital Medical University, Beijing, China

Address correspondence to Yuan Wu, wuyuan@icdc.cn.

Wenpeng Gu and Feng Liao contributed equally to this article. The author order was determined by alphabetical order.

The authors declare no conflict of interest.

See the funding table on p. 18.

*C*lostridioides difficile is a gram-positive anaerobic bacterium. Toxigenic *C. difficile* causes antibiotic-associated diarrhea, colitis, or severe lethal pseudomembranous enterocolitis in patients through the secretion of enterotoxin A (TcdA) and cytotoxin B (TcdB), referred to as *C. difficile* infection (CDI) (1, 2). In recent years, with the widespread use of antimicrobial drugs, the increased resistance of *C. difficile* and the emergence of highly pathogenic strains have led to increased levels of morbidity and mortality from CDI (3–5).

*C. difficile* infection has gained public health attention in China in recent years, and the most prevalent genotypes of *C. difficile* are sequence types (STs) 3, 35, and 54 (6, 7). In addition, the number of isolates from clade 4, especially ST37, is much greater in China than in other countries (7). Previously, we performed a retrospective study of community-acquired *C. difficile* infection in Southwest China from 2013 to 2016. ST35, ST54, and ST3 had the most distributed genotype profiles. The ST35 and ST3 strains were commonly found in children, but the ST54 strain was found in adults (8). Subsequently, we continued the surveillance and epidemiological investigation of *C. difficile* infection. Similar results were found in a previous study, but changes in the genotypes of the strains were identified. The ST37 isolates were gradually isolated in Southwest China and became the dominant strains (9).

In this study, we systematically analyzed the epidemiological features and pattern changes of community-acquired *C. difficile* in Southwest China for 7 years of data. In addition, the biological characteristics of some representative isolates were investigated to determine the reasons for these changes.

## RESULTS

### Epidemiological features and pattern changes

From 2013 to 2020, a total of 1,368 diarrhoeal fecal samples were collected from four sentinel hospitals in this study. Most of the patients were from departments of infectious medicine and pediatrics at sentinel hospitals, and the proportion of males was higher than that of females (Table 1). Children under 5 years of age with diarrhea accounted for 65.5% of all patients, followed by elderly individuals over 60 years of age (13.9%). A total of 53.9% of patients had diarrhea less than five times a day, and 80.5% of the feces were water (Table 1). A total of 184 of all the fecal samples were positive for the *tpi* gene, with a positivity rate of 13.45%. One hundred and sixty total fecal samples were positive for *tcdA* (11.7%), and 174 total fecal samples were positive for *tcdB* (12.7%). All the stool samples were negative for binary toxins. The *tcdB*+ gene in the feces was subsequently used as a positive indicator in the statistical analysis. Therefore, the *tcdB* positive rate in fecal samples of *C. difficile* in this study was 12.7%.

Sixty-nine *C. difficile* strains were isolated from all 1,368 diarrhea samples, and the isolation rate of the strains was 5.04%. The *C. difficile* isolation rate from the *tcdB*-positive feces was 39.7% (69/174). Among these strains, 59 (85.51%) were toxigenic (both *tcdA*+/*tcdB*+), and 10 (14.49%) were nontoxigenic (*tcdA*−/*tcdB*−). The binary toxins *cdtA* and *cdtB* were both negative.

Logistic regression was performed to analyze the *tcdB*+ gene in the fecal samples and strain isolation. The results showed that the risk factors for *tcdB*+ in the stool were a sentinel hospital (95% CI: 1.216–2.899) and fever (95% CI: 1.170–3.989). The univariate chi-square test also indicated statistical significance among sentinel hospitals ($\chi 2 = 10.76$, $P = 0.013$). Fecal samples from hospitals A (15.2%) and B (14.8%) were more positive for *tcdB* than those from hospital D (8.0%). No significant difference was found for the *C. difficile* strain isolation results according to logistic regression (Table 2).

The results of *C. difficile* coinfection with other pathogens were shown in Table 3. For all the diarrheal patients, 34 patients were coinfected with *tcdB*+ in stool samples, with more *rotavirus* (29.41%, 10/34) and diarrheagenic *Escherichia coli* (38.24%, 13/34) present. The number of cases of other pathogens coinfected with positive *C. difficile* isolates was 10 and was dominated by *rotavirus* (30.00%, 3/10), *adenovirus* (20.00%, 2/10), and diarrheagenic *E. coli* (30.00%, 3/10).

**TABLE 1** General information of the diarrhea patients included in this study

| Parameter | Factors | Case numbers | Proportion (%) |
|---|---|---|---|
| Sentinel hospitals | A | 336 | 24.6 |
| | B | 460 | 33.6 |
| | C | 221 | 16.1 |
| | D | 351 | 25.7 |
| Departments | Pediatric Clinic | 422 | 30.8 |
| | Infectious medicine | 444 | 32.5 |
| | Emergency medicine | 229 | 16.7 |
| | Others | 273 | 20 |
| Genders | Male | 816 | 59.6 |
| | Female | 552 | 40.4 |
| Age groups | ≤5 years | 896 | 65.5 |
| | 6–18 years | 49 | 3.6 |
| | 19–59 years | 233 | 17 |
| | ≥60 years | 190 | 13.9 |
| Occupations | Diaspora children | 753 | 55 |
| | Children in childcare | 114 | 8.3 |
| | Students | 46 | 3.4 |
| | Farmers | 90 | 6.6 |
| | Workers and others | 365 | 26.7 |
| Years | 2013 | 176 | 12.8 |
| | 2014 | 549 | 40.1 |
| | 2015 | 220 | 16.1 |
| | 2016 | 33 | 2.4 |
| | 2018 | 194 | 14.2 |
| | 2019 | 146 | 10.7 |
| | 2020 | 50 | 3.7 |
| Diarrhea days before hospitals | ≤5 days | 1,034 | 75.6 |
| | 5–10 days | 146 | 10.7 |
| | ≥10 days | 188 | 13.7 |
| Diarrhea times/per day | ≤5 times | 737 | 53.9 |
| | 5–10 times | 521 | 38.1 |
| | ≥10 times | 110 | 8 |
| Fecal property | Water stool | 1,101 | 80.5 |
| | Mucoid stool | 160 | 11.7 |
| | Blood stool | 107 | 7.8 |
| Fever | Yes | 414 | 30.3 |
| | No | 954 | 69.7 |
| Vomit | Yes | 418 | 30.6 |
| | No | 950 | 69.4 |
| Vomit days before hospitals | ≤5 days | 408 | 97.6 |
| | 5–10 days | 6 | 1.4 |
| | ≥10 days | 4 | 1 |
| Vomit times/per day | ≤5 times | 349 | 83.5 |
| | 5–10 times | 69 | 16.5 |
| Use antibiotics before hospitals | Yes | 74 | 5.4 |
| | No | 1,294 | 94.6 |

Interestingly, the number of coinfected patients under 5 years of age accounted for 91.18% (31/34) of the total cases, as shown in Table 3. Among children under 5 years of age, those with CDI coinfections were also predominantly *rotavirus* and diarrheagenic *E. coli*, both in terms of fecal *tcdB+* and *C. difficile* isolation cultures.

There were 17 STs generated among the 69 isolates in this study, showing a high degree of discrete features, as shown in Fig. 1a. ST3 (21.74%), ST35 (17.39%), ST54

**TABLE 2** Logistic regression of clinical features for *tcdB* gene detection and isolation

| Factors | *tcdB* gene results of fecal samples | | | | Isolation results | | | |
|---|---|---|---|---|---|---|---|---|
| | OR | 95% C.I. | | *P* value | OR | 95% C.I. | | *P* value |
| | | Lower | Upper | | | Lower | Upper | |
| Sentinel hospitals | 1.878 | 1.216 | 2.899 | 0.004 | 1.546 | 0.832 | 2.871 | 0.168 |
| Departments | 0.804 | 0.498 | 1.298 | 0.373 | 0.991 | 0.482 | 2.038 | 0.981 |
| Genders | 1.320 | 0.717 | 2.431 | 0.373 | 1.102 | 0.506 | 2.399 | 0.808 |
| Age groups | 1.373 | 0.701 | 2.691 | 0.356 | 0.470 | 0.132 | 1.667 | 0.242 |
| Occupations | 0.842 | 0.561 | 1.264 | 0.407 | 1.862 | 0.741 | 4.678 | 0.186 |
| Years | 0.899 | 0.746 | 1.084 | 0.265 | 0.899 | 0.690 | 1.172 | 0.432 |
| Diarrhea days before hospitals | 1.425 | 0.755 | 2.687 | 0.274 | 0.725 | 0.381 | 1.381 | 0.328 |
| Diarrhea times/per day | 1.083 | 0.691 | 1.697 | 0.729 | 1.566 | 0.845 | 2.903 | 0.154 |
| Fecal property | 1.043 | 0.405 | 2.687 | 0.930 | 1.210 | 0.305 | 4.801 | 0.786 |
| Fever | 2.160 | 1.170 | 3.989 | 0.014 | 1.349 | 0.613 | 2.969 | 0.457 |
| Vomit days before hospitals | 0.506 | 0.180 | 1.419 | 0.195 | 1.201 | 0.327 | 2.622 | 0.998 |
| Vomit times/per day | 0.816 | 0.391 | 1.703 | 0.588 | 0.632 | 0.245 | 1.630 | 0.343 |
| Use antibiotics before hospitals | 0.989 | 0.478 | 2.043 | 0.975 | 0.857 | 0.295 | 2.487 | 0.776 |

(13.04%), ST2 (7.25%), ST39 (7.25%), and ST37 (7.25%) were the most common STs of *C. difficile* (Fig. 1a). Two clades of isolates based on STs were found, namely, clade 1 and clade 4, and a great diversity of STs of the strains were identified in clade 1. The phylogenetic tree based on the genome of *C. difficile* was also divided into two clusters corresponding to clade 1 and clade 4, as shown in the yellow and green areas, respectively, in Fig. 1b.

Through the surveillance data, we found that the *C. difficile* strains in Southwest China exhibited genotype changes. As shown in Fig. 1c, three major STs (ST3, ST35, and ST54) were isolated from 2013 to 2016, and no ST37 genotype was found in this period. From 2018 to 2020, the ST37 strain became the dominant *C. difficile* in this period, while the number of isolates of the ST3, ST35, and ST54 strains decreased correspondingly.

## Antibiotic resistance

All 69 *C. difficile* strains were sensitive to metronidazole, amoxicillin/clavulanic acid, and vancomycin, as shown in Table 4. A total of 87.00% of the *C. difficile* strains were resistant to erythromycin, followed by gentamicin (85.50%). The rates of antibiotic resistance to clindamycin were 63.80%, 37.70% for ceftazidime, 27.50% for imipenem, 21.70% for cefotaxime, and 13.00% for ciprofloxacin (Table 4). The MIC values of the antibiotic susceptibility tests for all strains were shown in Table S1.

The statistical results revealed that the resistance rates of *C. difficile* strains to IPM (*P* = 0.010), CTX (*P* = 0.019), CAZ (*P* = 0.000), and CIP (*P* = 0.000) were significantly different between clade 1 and clade 4. Most of the isolates resistant to IPM, CTX, CAZ, and CIP were clade 4 strains, as shown in Fig. 2a through d. Only the CIP resistance rate and ST type of *C. difficile* were significantly different (*P* = 0.032). Among these isolates, the ST37 and ST39 strains belonged to the clade 4 group of *C. difficile* (Fig. 2e).

**TABLE 3** Coinfection with other enteric pathogens for CDI patients

| | *C. difficile* infection | Positive for any virus (case numbers) | | | | Positive for any bacteria (case numbers) | | Cases of coinfection |
|---|---|---|---|---|---|---|---|---|
| | | Rotavirus | Norovirus | Adenovirus | Astrovirus | Diarrheagenic *E. coli* | *Salmonella* | |
| Total cases with diarrhea | Fecal samples of *tcdB*+ | 10 | 5 | 4 | 1 | 13 | 1 | 34 |
| | *C. difficile* isolation+ | 3 | 1 | 2 | 0 | 3 | 1 | 10 |
| Children under 5 years old | Fecal samples of *tcdB*+ | 10 | 5 | 2 | 1 | 12 | 1 | 31 |
| | *C. difficile* isolation+ | 3 | 1 | 1 | 0 | 3 | 1 | 9 |

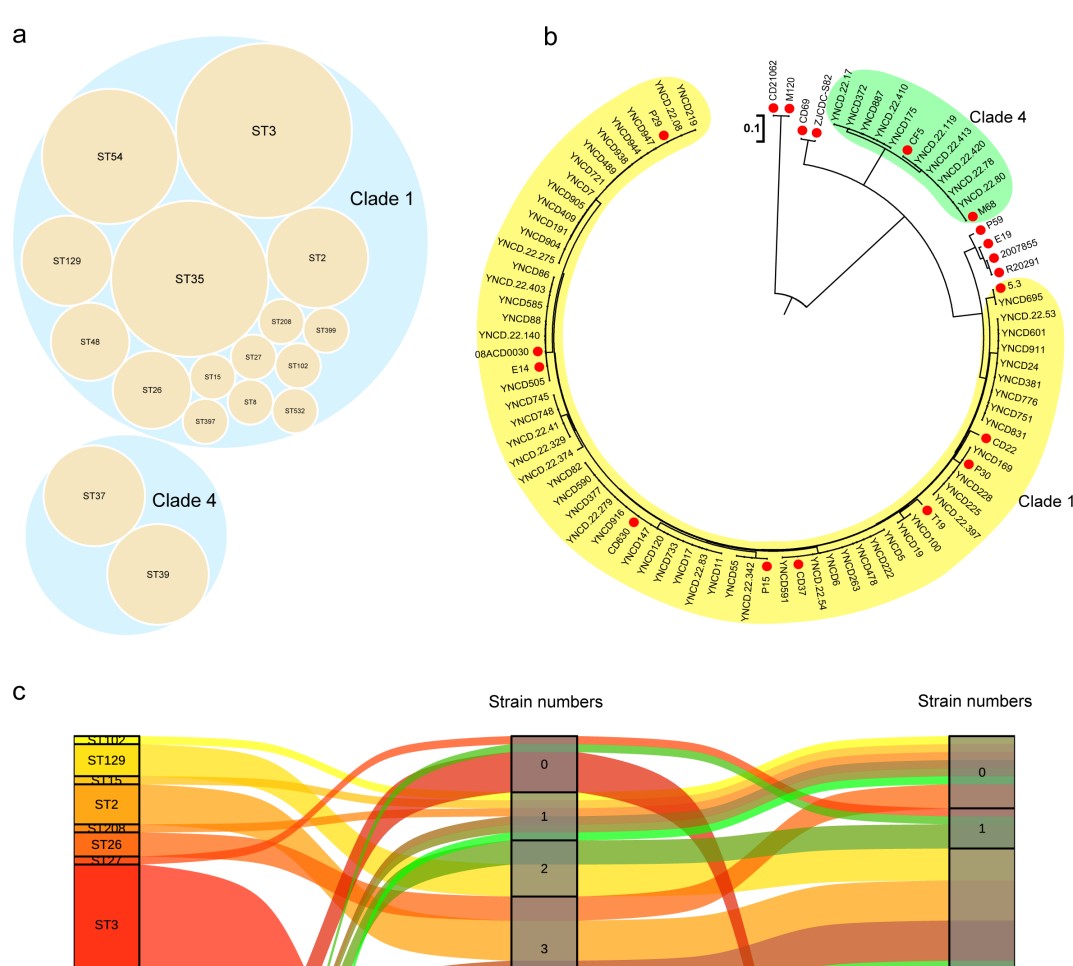

**FIG 1** The distribution of STs, phylogenetic tree of the core genome and pattern changes of *C. difficile* in this study. (a) Distribution of STs of all *C. difficile* isolates; (b) phylogenetic tree of *C. difficile* based on the core genome, where the yellow area represented clade 1, the green area represented clade 4, and the red dot represented the reference genome; (c) Sankey diagram of *C. difficile* genotype pattern changes.

## Genomic comparison

The results of the genome collinearity analysis revealed that, compared with R20291, CD413 underwent substantial gene rearrangement, as shown in Fig. 3a. Gene ectopics and inversions are commonly found in CD413 *C. difficile*, with the majority of genes located in the reverse complementary region compared with the reference region. Gene rearrangements were also very significant in the CD413 strain compared with the CD21062 reference strain, as shown in Fig. 3b. Large swaps and inversions of gene positions can also be observed, even to a greater extent than in the R20291 reference genome.

**TABLE 4** The antibiotic resistance of the *C. difficile* strains in this study

| Antibiotics | Sensitive | % | Intermediate | % | Resistant | % |
|---|---|---|---|---|---|---|
| MTZ | 69 | 100.00% | 0 | 0 | 0 | 0 |
| E | 6 | 8.70% | 3 | 4.30% | 60 | 87.00% |
| AMC | 69 | 100.00% | 0 | 0 | 0 | 0 |
| CIP | 29 | 42.00% | 31 | 44.90% | 9 | 13.00% |
| IPM | 48 | 69.60% | 2 | 2.90% | 19 | 27.50% |
| TE | 63 | 91.30% | 5 | 7.20% | 1 | 1.40% |
| CN | 2 | 2.90% | 8 | 11.60% | 59 | 85.50% |
| DA | 19 | 27.50% | 6 | 8.70% | 44 | 63.80% |
| VA | 69 | 100.00% | 0 | 0 | 0 | 0 |
| CAZ | 13 | 18.80% | 30 | 43.50% | 26 | 37.70% |
| AML | 65 | 94.00% | 4 | 5.80% | 0 | 0 |
| CTX | 0 | 0 | 54 | 78.30% | 15 | 21.70% |

Compared with R20291, a total of 61,589 SNP variants were detected in CD413, of which 48,049 were transitions and 13,540 were transversions, with a Ti/Tv rate of 3.55. Most of the variations were located in the upstream or downstream regions of the gene, accounting for 43.91% and 45.24%, respectively. The proportion of variants occurring in the gene coding region was 8.85%, as shown in Fig. 4a. The variant locations of the InDels were also predominantly located upstream of the gene (43.58%) and downstream of the gene (45.54%), whereas the gene coding region accounted for only 3.92%, as shown in Fig. 4b. The functional annotations of the SNPs and InDels indicated that the genome functional variants were involved mainly in the metabolism, genetic information, and environmental information processing of the strain. Among them, carbohydrate metabolism, amino acid metabolism, and nucleotide metabolism were dominant, as shown in Fig. 4c and d.

Compared with those in CD21062, a total of 101,900 SNP variants were detected in the CD413 strain, of which 79,180 were transitions and 22,720 were transversions, with a Ti/Tv ratio of 3.48. Most of the variant locations were also located in the upstream or downstream regions of the gene, accounting for 44.34% and 45.57%, respectively. The proportion of mutations in the gene coding region was 8.36%, as shown in Fig. 4e. The mutation locations of the InDels were also predominantly located in the upstream (43.64%) and downstream (46.98%) regions of genes, whereas the gene coding region accounted for only 2.21%, as shown in Fig. 4f. The results of the functional annotation of the SNPs and InDels indicated that the genome functional variations were involved mainly in the metabolism of the strain, genetic information, and environmental information processing. Among them, carbohydrate metabolism, amino acid metabolism, metabolism of cofactors, and vitamins and nucleotide metabolism were the main functions, as shown in Fig. 4g and h .

We also compared the mobile genetic elements (MGEs) of these representative strains, and the results are shown in Table 5. Among the four representative strains, CD413 had all the MGE elements with insertion sequences, transposons, plasmids, phages, and integrative and conjugative elements, most of which were associated with antibiotic resistance and virulence factors of the strain. For example, the transposon Tn925 is associated with tetracycline resistance in the strain; the plasmid DOp1 (repUS43) is associated with tetracycline and β-lactam antibiotic resistance in the strain; and the prophage is associated with extracellular enzymes and adhesion in the strain.

## RNA-seq and verification of the results

Principal component analysis (PCA) of four representative strains revealed that the R20291 and CD413 strains had close correlation distributions (Fig. 5a), while CD21062 and CD279 were relatively distant from each other. The pairwise comparison of the differentially expressed genes (DEGs) of these isolates revealed that several genes were

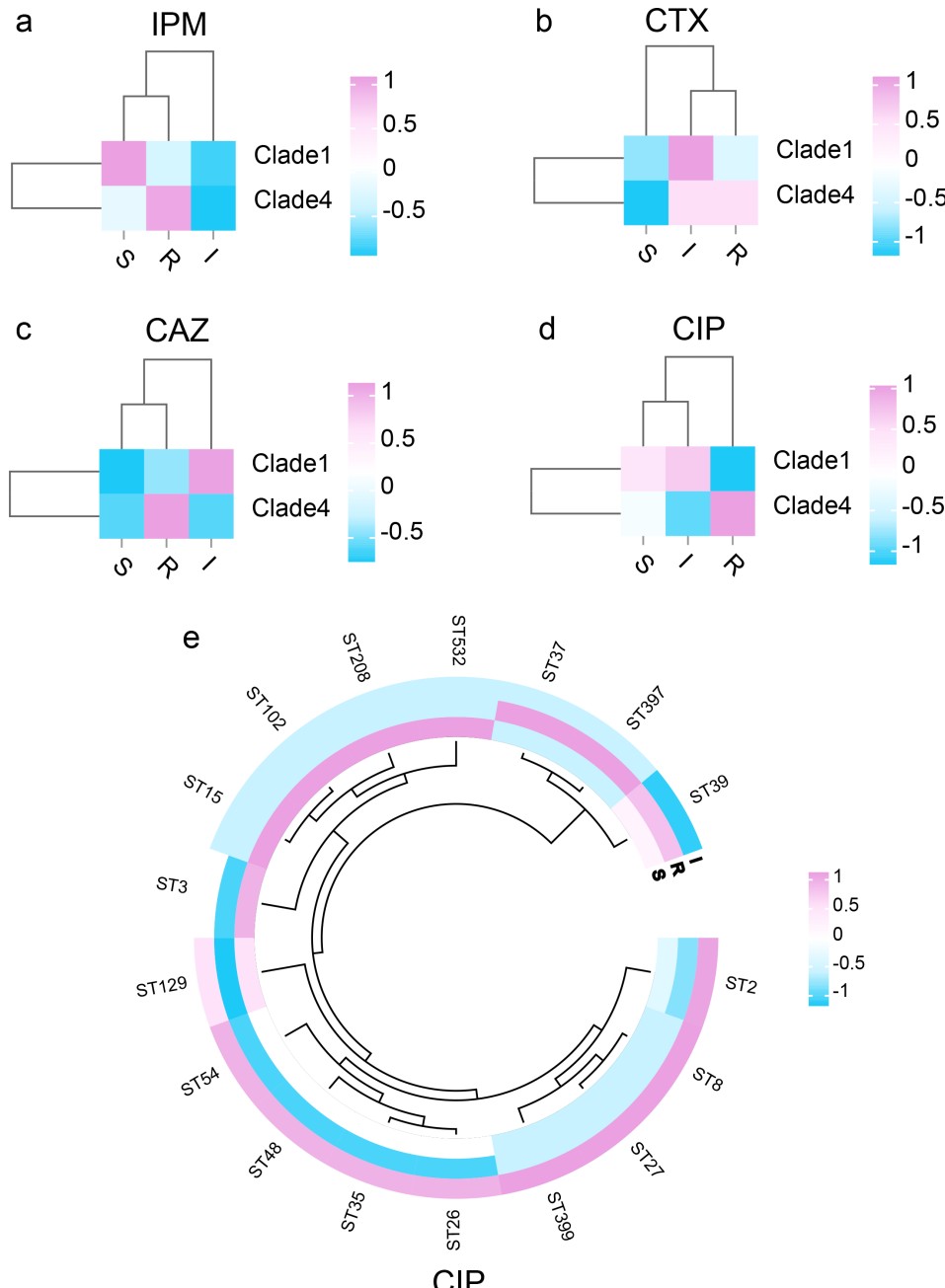

**FIG 2** Heatmaps of the statistical significance of antibiotics for clades and STs. (a) The clustering heatmap of IPM resistance results with clades of strains; (b) the clustering heatmap of CTX resistance results with clades of strains; (c) the clustering heatmap of CAZ resistance results with clades of strains; (d) the clustering heatmap of CIP resistance results with clades of strains; (e) the clustering heatmap of CIP resistance results with STs.

upregulated or downregulated. The Venn diagram showed that 99 core gene sets were identified among the pairwise comparisons of the four *C. difficile* strains; the lowest number of unique gene sets was found in the CD413 vs R20291 group, and the highest number of unique genes was found in the R20291 vs CD21062 group, as shown in Fig. 5b.

KEGG enrichment revealed that the genes upregulated in the CD279 strain were mainly related to the ABC transporter, sugar, and amino acid metabolism pathways in the CD279 vs R20291 comparison (Fig. 5c). In the CD279 vs CD21062 comparison, the genes

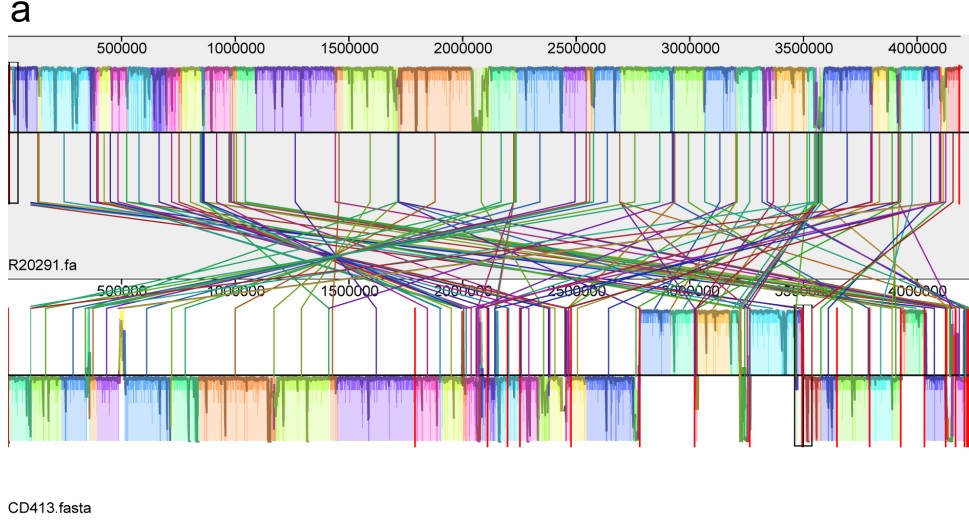

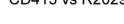

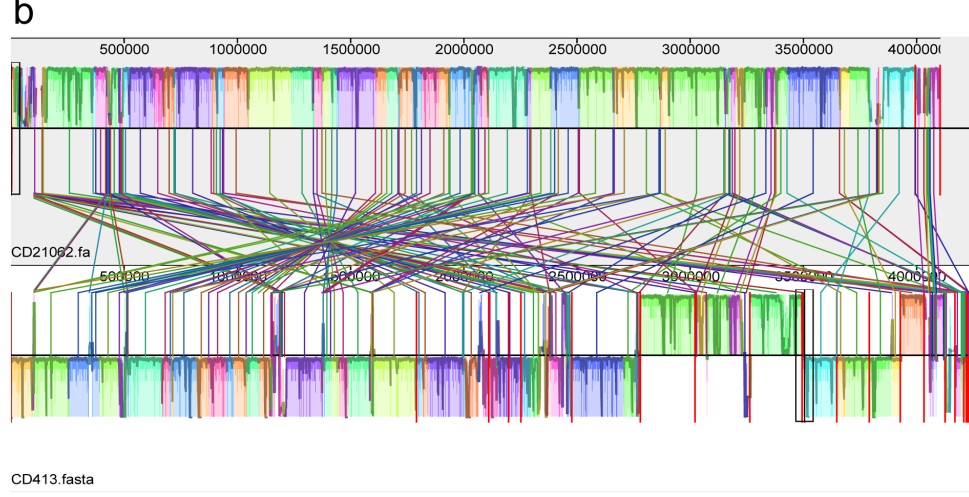

FIG 3 Collinearity analysis between the CD413 and R20291 and CD21062 reference genomes. (a) Genomic collinearity analysis between CD413 and R20291; (b) genomic collinearity analysis between CD413 and CD21062. The blocks of the same color represent homologous genes, which are connected via color line segments; color blocks above the centerline represent genomic sequences in the forward orientation; and blocks below the centerline indicate alignment in the reverse complementary orientation.

upregulated in CD279 were enriched mainly in the two-component system and quorum sensing (QS) pathways of the strains (Fig. 5d). The genes upregulated in CD413 were mainly enriched in the phosphotransferase system (PTS), ribosome, and some sugar metabolism pathways of the strain in the CD413 vs CD279 comparison (Fig. 5e).

To validate the RNA-seq results, RT-qPCR of mRNA from strains cultured for 24 h was used to analyze the *nisR* and *RS16530* genes. The *nisR* gene is involved in two-component systems and QS pathways related to the synthesis of antimicrobial peptides and bacterial immunity. The *RS16530* gene encodes PTS fructose transporter subunit II C, which is involved in the PTS pathway for the transport of fructose in *C. difficile*. The relative expression of *nisR* indicated that the CD413 strain had the highest expression level among the four strains, followed by CD279, and that of R20291 was the lowest. ANOVA showed that, except for the CD21062 and CD279 groups, the pairwise comparisons of the four strains were statistically significant ($P < 0.05$) (Fig. 5f). The relative expression of

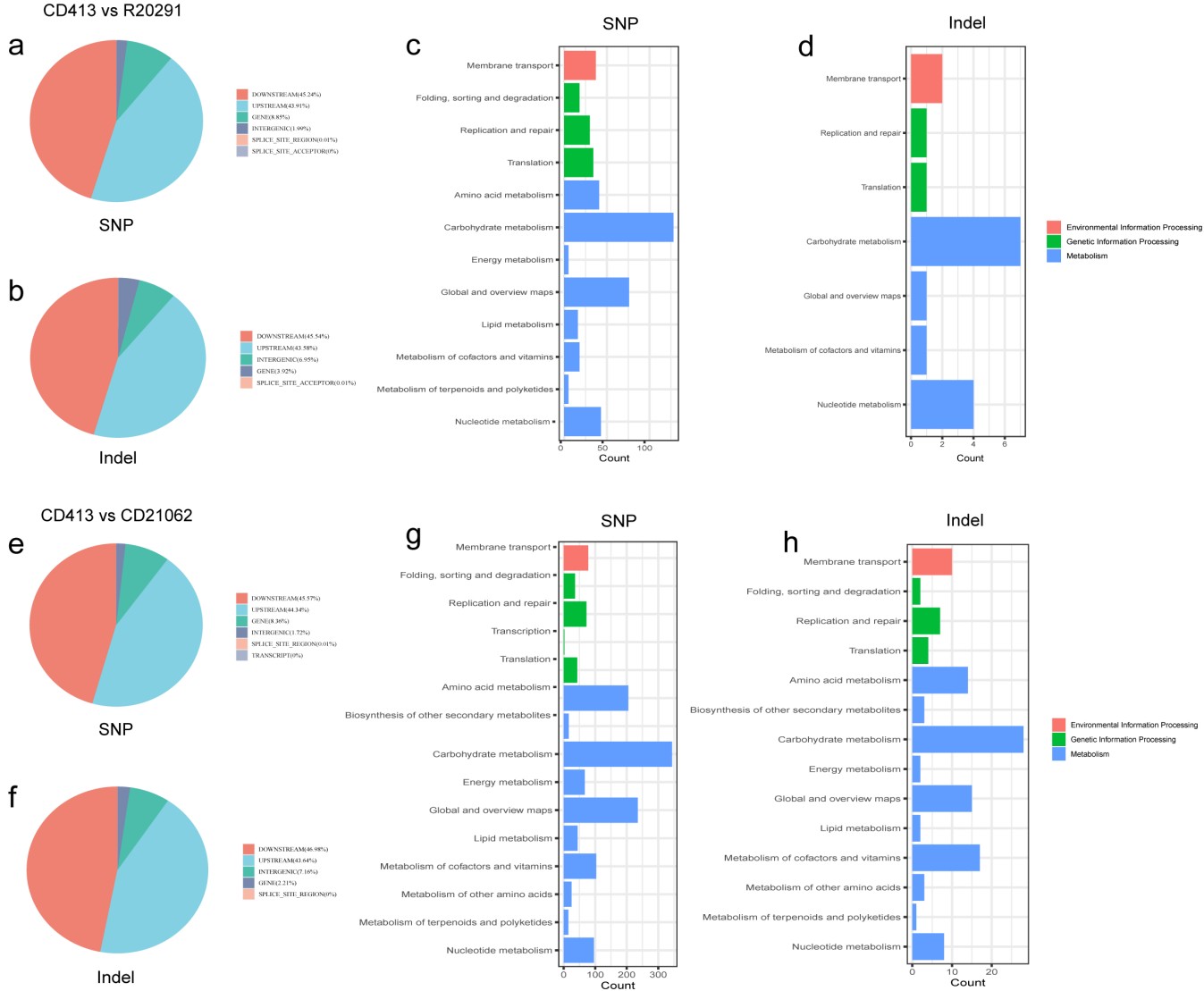

**FIG 4** Variation and annotation analysis of CD413 compared with the R20291 and CD21062 reference genomes. (a) Pie chart of SNP annotation results of CD413 vs R20291; (b) pie chart of InDel annotation results of CD413 vs R20291; (c) statistical graph of classification of SNP functional annotations of CD413 vs R20291; (d) statistical graph of classification of InDel functional annotations of CD413 vs R20291; (e) pie chart of SNP annotation results of CD413 vs CD21062; (f) pie chart of InDel annotation results of CD413 vs CD21062; (g) statistical graph of classification of SNP functional annotations of CD413 vs CD21062; (h) statistical graph of classification of InDel functional annotations of CD413 vs CD21062.

*RS16530* showed that the CD413 strain had the highest expression level at the 24 h time point, and R20291 had the lowest (Fig. 5g). Except for the CD21062 vs CD279 group, pairwise comparisons of the four strains revealed significant differences ($P < 0.05$). The RT-qPCR and RNA-seq results were consistent.

## Biofilm assay

The biofilm formation assay showed that obvious bacterial clumps and biofilms could be observed after CV staining of the four strains (Fig. 6a through d). Among them, the CD279 strain showed the greatest density of bacterial biofilms under light microscopy.

The absorbance measurements showed that the CD279 strain had the strongest biofilm formation ability, followed by R20291 and CD21062, while CD413 had the lowest biofilm formation ability (Fig. 6e). ANOVA indicated significant differences ($P < 0.05$) in the OD values among the four strains for pairwise comparisons.

**TABLE 5** Comparison of mobile genetic elements between representative strains

| Strains | MGE type | Numbers | Element type |
|---------|----------|---------|--------------|
| R20291 | IS | 22 | IS200/IS605; IS256; IS3 |
| | Transposon | 1 | Tn6216 |
| | Plasmid | 0 | -[a] |
| | Phage | 9 | - |
| | ICE | 0 | - |
| CD21062 | IS | 35 | IS200/IS605 |
| | Transposon | 1 | Tn6216 |
| | Plasmid | 0 | - |
| | Phage | 7 | - |
| | ICE | 1 | Tn6009 |
| CD413 | IS | 12 | IS200/IS605; IS3; IS1182; IS256 |
| | Transposon | 1 | Tn925 |
| | Plasmid | 1 | DOp1(repUS43) |
| | Phage | 13 | - |
| | ICE | 2 | CTn5; Tn6009 |

[a] "–" indicates the absence of this type of genetic element.

## Toxin gene expression and cytotoxicity

The relative expression of *tcdA* in R20291 and CD21062 was higher than that in CD279 and CD413 of the *C. difficile* strains (Fig. 7a) ($P < 0.05$). However, the differences between the R20291 vs CD21062 group and the CD279 vs CD413 group were not significant. The R20291 strain showed the highest *tcdB* relative expression level, followed by that of CD21062 and that of CD279 was the lowest (Fig. 7b). Pairwise comparisons of the four representative strains of *tcdB* revealed significant differences ($P < 0.05$).

The cultures of *C. difficile* strains harboring TcdA showed that R20291 had the highest concentration of the toxin TcdA, which was significantly different from that of other representative strains ($P < 0.05$). There was no significant difference in the TcdA toxin concentrations among CD21062, CD279, and CD413 (Fig. 7c). Similarly, R20291 had the highest concentration of TcdB toxin, which was significantly different from that of the other three strains ($P < 0.05$). The TcdB concentration in CD21062 cells was significantly different from that in CD413, and no significant difference was found between CD279 and CD413 strains, as shown in Fig. 7d.

The R20291 strain exhibited apparent cytopathic effects under light microscopy after infection of Hep-2 cells. Most of the cells were detached, deformed, and rounded after infection (Fig. 7e). However, Hep-2 cells infected with the CD21062, CD279, and CD413 strains did not exhibit significant cytopathic effects. Normal cell morphology and structure could be observed, and the cells were not detached or deformed (Fig. 7f). The LDH release assay showed that the LDH content of Hep-2 cells after R20291 infection was significantly higher than that of other representative strains ($P < 0.05$) (Fig. 7g). The LDH levels of Hep-2 cells infected with CD21062, CD279, and CD413 were not significantly different according to pairwise comparisons.

The results of the cell viability assay revealed that the average cell survival rates after the interaction of R20291, CD21062, CD279, and CD413 with Hep-2 cells were 17.00% ± 1.49%, 67.33% ± 2.34%, 64.89% ± 2.33%, and 63.44%± 1.83%, respectively. Hep-2 cells had the lowest cell survival rate after infection with the R20291 strain, and the difference was statistically significant from that of the remaining three strains, as shown in Fig. 7h. No significant difference was found in the cell survival rate after interaction of the CD21062, CD279, and CD413 strains with cells.

## DISCUSSION

This study aimed to perform an epidemiological study of *C. difficile* infections in the diarrhea population of Southwest China, so we included diarrhea samples from the

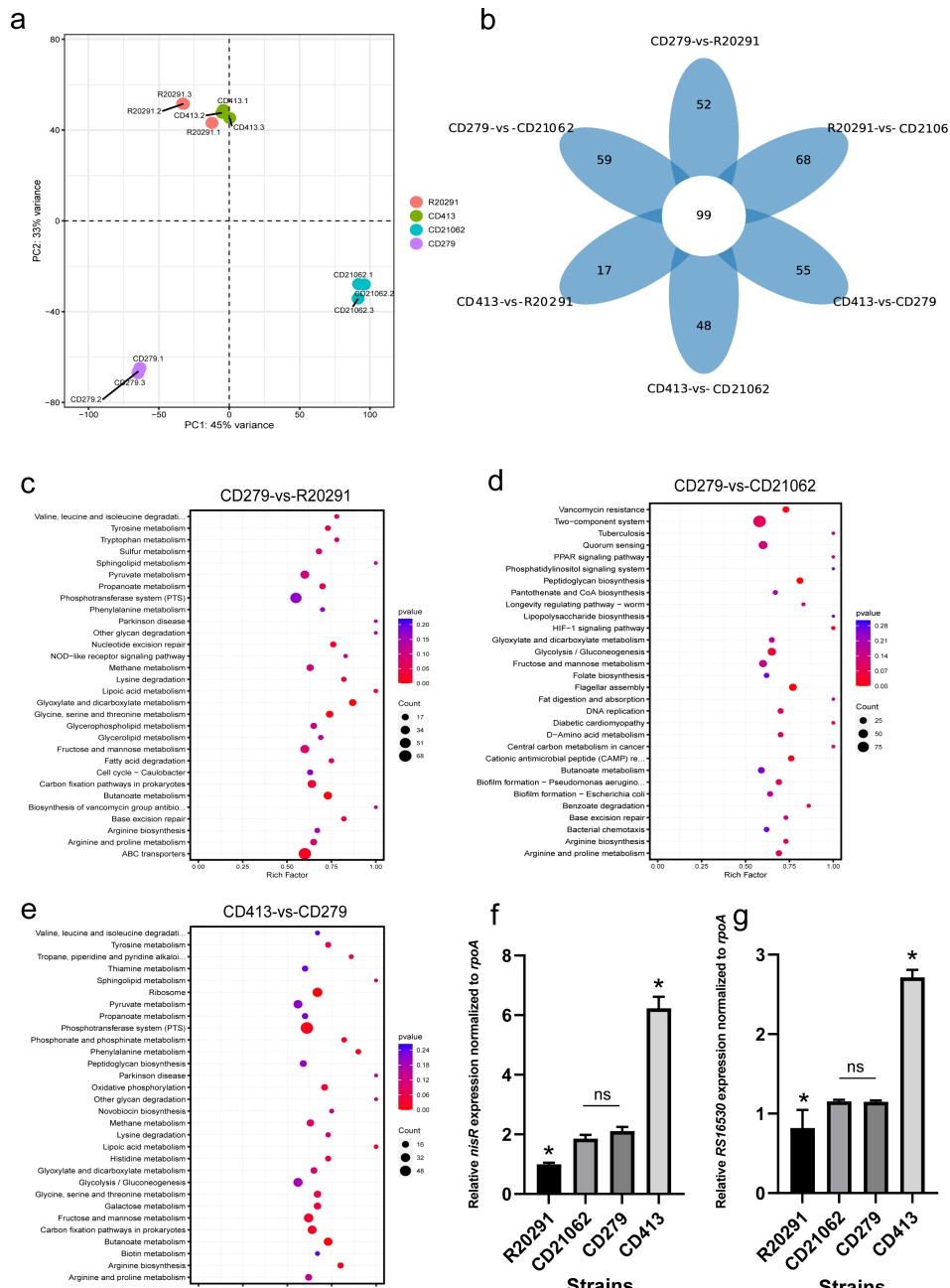

**FIG 5** RNA-seq of four STs of *C. difficile* strains after bacterial culture. (a) Principal component analysis (PCA) of four representative strains; (b) venn diagram of the differentially expressed genes of *C. difficile*; (c) KEGG pathway enrichment results of the CD279 vs R20291 group; (d) KEGG pathway enrichment results of the CD279 vs CD21062 group; (e) KEGG pathway enrichment results for the CD413 vs CD279 group; (f) relative expression levels of the *nisR* gene in four representative strains; (g) relative expression levels of the *RS16530* gene in four representative strains.

whole population. However, such a large group (65.5%) of children patients under the age of 5 in this study, led to the low incidence of CDI and only 5.04% of lifetime prevalence because of the well-known data in the literature about the CDI among young children (10, 11).

Several studies have reported the molecular epidemiology of *C. difficile* in China (5, 6, 12, 13). In general, most clinical *C. difficile* isolates in China belong to ST3, ST35, and ST54 of clade 1, but the proportion of *C. difficile* isolates from clade 4, especially ST37, is much

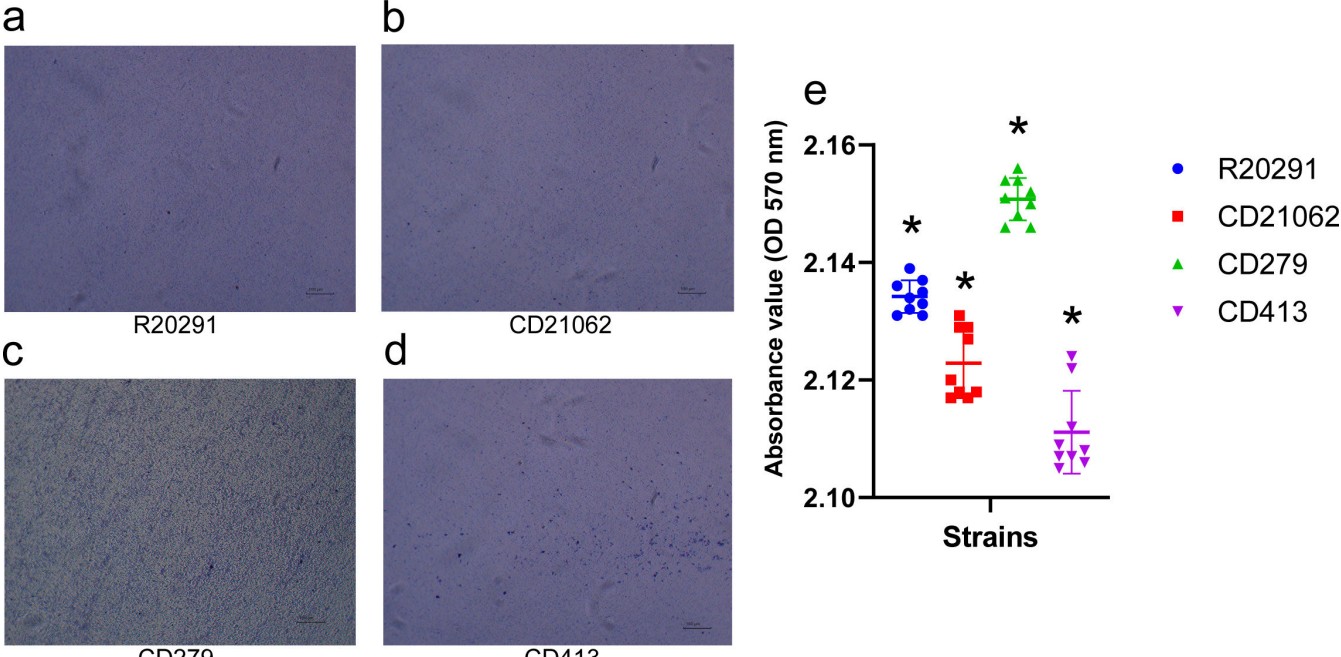

**FIG 6** The biofilm formation of the four STs of *C. difficile* in this study. (a) R20291 biofilm formation observed under a light microscope (10×); (b) CD21062 biofilm formation observed under a light microscope (10×); (c) CD279 biofilm formation observed under a light microscope (10×); (d) CD413 biofilm formation observed under a light microscope (10×); (e) Comparison of the OD values in the biofilm formation assay of the four representative strains.

greater than that in other regions. Our study also revealed that the genotypes of community-acquired *C. difficile* infection strains in Southwest China were also dominated by ST3, ST35, and ST54, while the ST37 genotype strain accounted for a certain proportion of the isolates. However, based on 7 years of surveillance data, we found that the genotypes of *C. difficile* in this area underwent pattern changes. No ST37 genotype strains were isolated from 2013 to 2016. However, after 2018, *C. difficile* ST37 was gradually isolated and became the dominant genotype during this period. The proportions of ST3, ST35, and ST54 gradually decreased.

Changes and variations in the genotypes of many pathogens have been reported (14–17), and the reasons for these changes might be related to their genomic or biological characteristics, as well as host or environmental factors (18, 19). Xu et al. (20) reported that the ST37 strains from China were divided into four distinct sublineages based on genomic sequencing, which revealed five importations and international sources. KEGG analysis revealed that 10 metabolic pathways were significantly enriched in the mutations among isolates associated with severe CDI, and these mutations were involved in glycol metabolism, amino acid metabolism, and biosynthesis. In this study, we wanted to explore whether the ST37 genotype strains became dominant in terms of their biological features. Therefore, the highly virulent *C. difficile* R20291 (RT027) and the ST11 strain CD21062 (RT078) were both selected as representative strains (21, 22), and the ST54 and ST37 strains isolated from this area were also included. Because of the type changing of strains, the selection of representative strains for biological studies was a key point in this study. It should be noted that ST35 and ST3 were more prevalent in children, while ST54 was the most prevalent in adults with diarrhea in Southwest China. Current studies suggest that intestinal *C. difficile* in children may be associated with colonization, whereas most studies have focused more on adult *C. difficile* infections. For this reason, we selected ST54 adult isolate of the *C. difficile* strain as a representative to be analyzed simultaneously with ST37, which was a type-changing strain, in an attempt to reveal the reasons for this switch in terms of biological characteristics. For genomic comparison, the *C. difficile* strain CD413 (ST37 genotype) underwent a high degree of genetic

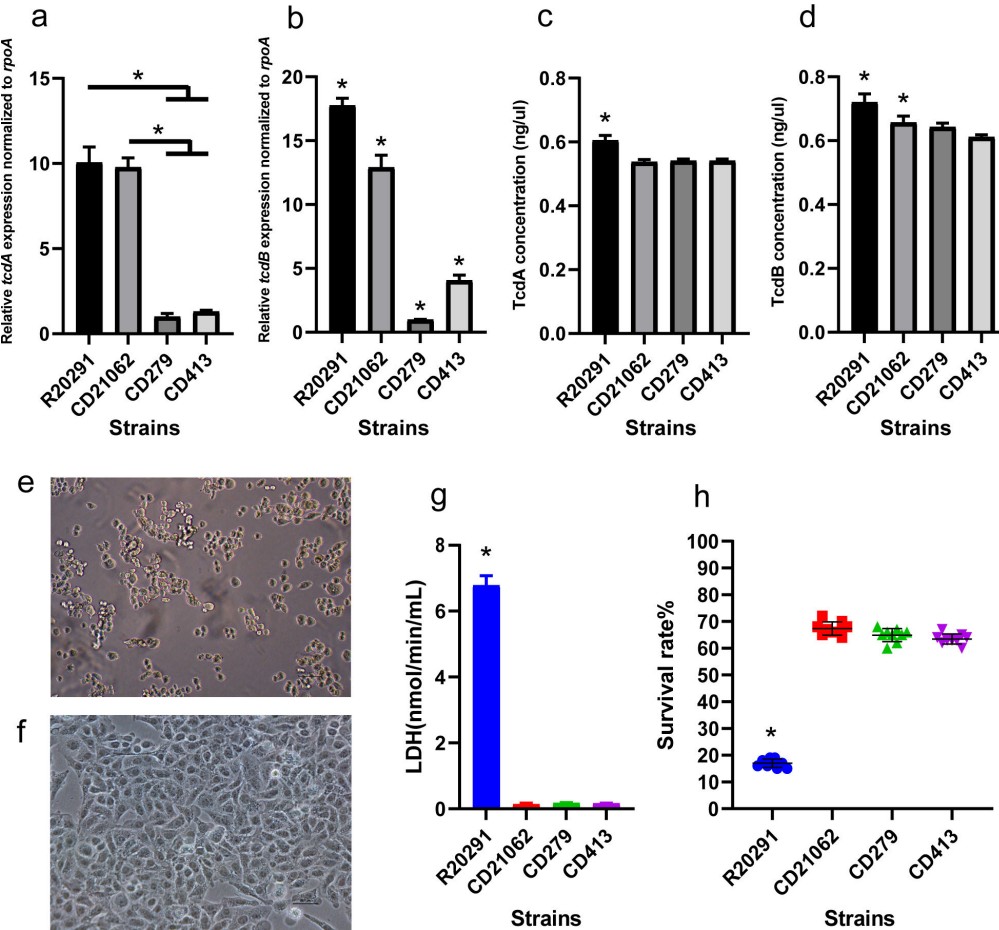

**FIG 7** The toxin expression of four ST strains and their cytotoxicity to Hep-2 cells. (a) Relative expression levels of *tcdA* gene in representative strains at 24 h; (b) relative expression levels of *tcdB* gene in four representative strains at 24 h; (c) quantification of the TcdA toxin among the four representative strains at 24 h; (d) quantification of the TcdB toxin among the four representative strains at 24 h; (e) light microscopy observation of R20291 after 3 h of interaction with Hep-2 cells (10×); (f) CD21061, CD279, and CD413 after interactions with Hep-2 cells by light microscopy observation (10×); (g) the absorbance measurement of the LDH concentration among the four representative strains infected with cells; (h) survival rates of four representative *C. difficile* strains after interaction with Hep-2 cells.

rearrangement. The SNP and InDel regions of the genomic variation were concentrated in the upstream or downstream regions of the coding genes, and these regions were closely related to the gene expression regulation of the bacteria. KEGG functional annotation revealed that these variations were involved mainly in the metabolism of the strain, carbohydrate metabolism, amino acid metabolism, nucleotide metabolism, etc. MGEs analyses revealed that CD413 had a high occurrence of gene transfer events and that these genetic elements conferred antibiotic resistance and some virulence factors to the strain. All these changes in genomic regions may contribute to its enhanced adaptability and transmission advantage to the environment.

The RNA-seq results showed that the transcriptomes of the ST37 strain and RT027 *C. difficile* were similar according to PCA. In addition, compared with those in the RT027 and RT078 strains, the genes highly expressed in *C. difficile* ST54 were enriched mainly in ABC transporters, two-component systems, quorum sensing, and sugar and amino acid metabolism pathways. ABC transporters function as molecular machines by coupling ATP binding, hydrolysis, and phosphate release to translocate diverse substrates across membranes through processes such as the high-affinity uptake of micronutrients into bacteria and the export of cytotoxic compounds from cells (23). Two-component systems

are ubiquitous signaling mechanisms in bacteria that enable intracellular changes from extracellular ones. These enriched pathways were critically related to bacterial nutrition, metabolism, immunity, virulence, and gene expression regulation (24). Recurrent *C. difficile* infections are frequently associated with the formation of biofilms regulated by quorum sensing, which is an interbacterial communication system usually composed of two-component systems that activate signal transduction pathways through interactions with receptors (25). Compared with the highly virulent strains in this study, the QS pathway of *C. difficile* ST54 was significantly enriched. These findings suggested that the transcriptome results were consistent with the bacterial biofilm phenotype results.

The ST37 genotype strain was enriched in the PTS, ribosome, and several sugar and amino acid metabolic pathways compared with the ST54 strain in this study. The bacterial phosphotransferase system carries out both catalytic and regulatory functions. It catalyses the transport and phosphorylation of a number of sugars and derivatives and carries out numerous regulatory functions related to carbon, nitrogen, and phosphate metabolism, chemotaxis, and the virulence of some pathogens (26). We considered that the pattern changes of strains might be closely related to the adaptability of the environment or ecological niches through the regulation of gene expression. An *in vitro* growth curve study of *C. difficile* (our unpublished data) revealed that CD413 *C. difficile* (ST37) rapidly reached the logarithmic growth phase after 12 h of culture, while the R20291 strain did not reach this phase after 24 h. This result might reflect the high metabolic status of the *C. difficile* ST37 genotype strain, with high levels of ribosome synthesis and sugar and amino acid metabolism.

From the results, gene transcription and protein expression of both toxins were highest in the R20291 strain, followed by CD21062, while the expression levels of the remaining two strains were similar. Although some statistical differences were found between the qPCR and ELISA method, the mRNA transcription and protein expression trends of the two toxins in the four *C. difficile* strains were consistent. The highly virulent R20291 strain showed the highest toxin expression levels and greatest cytotoxicity among the four strains. The cells exhibited cytopathic effects in a short time, and a large amount of LDH was released from the cells. Although the RT078 strain is also considered a highly virulent *C. difficile* strain, in the cell infection model in this study, its cytopathic effect was not as strong as that of RT027 on *C. difficile*. Our previous study of an animal infection model for different ST11 strains, including the RT078 strain (CD21062), showed that no animal died during the experimental procedure (27). Accordingly, we also considered that *C. difficile* strains with low virulence might be better adapted to the host or environment, resulting in persistent infection.

In China, the antimicrobial resistance profile of *C. difficile* has increased to erythromycin, clindamycin, and fluoroquinolones (13, 28). Clade 4 has been reported to demonstrate diversity in mobile genetic elements, and a high rate of MDR was found within clade 4, especially for the ST37 genotype (7). The rates of antibiotic resistance of community-acquired *C. difficile* strains in Southwest China were similar to those in other areas of China; specifically, clade 4 isolates were more resistant to antibiotics than clade 1 strains, and ST37 strains were more resistant to quinolones. Strengthening surveillance of the ST37 genotype of *C. difficile* is currently a critical point for *C. difficile* infection control. Moreover, for antibiotics with high resistance rates (such as erythromycin and gentamicin) in Southwest China, the impact of these resistance situations on clinical treatment options should be concerned. Whether there are specific factors in the local medical environment that contribute to the increase in resistance (such as antibiotic usage habits, or hospital infection control measures) need further investigation.

## MATERIALS AND METHODS

### General information

The case definition was diarrhea occurring more than three times/day, accompanied by changes in fecal traits. Community-acquired diarrhea was defined as the onset of diarrhea symptoms, while the patient was outside a healthcare facility or had no prior stay in a healthcare facility within the 12 weeks prior to symptom onset.

Four sentinel hospitals in Yunnan Province, Southwest China, were involved from 2013 to 2020, and 1,368 cases were corrected during that period. These four hospitals covered the entire Kunming area and were named A, B, C, and D. Stool samples from diarrhea patients were collected and transported to the Yunnan Provincial Center for Disease Control and Prevention for laboratory detection, and only the first stool specimen from each patient was included.

### *tcdB* gene detection in fecal samples

The genomic DNA of the fecal samples was extracted by using a fecal sample DNA extraction kit (Tiangen, Beijing) according to the manufacturer's instructions. All the DNA samples were tested for the virulence genes *tcdA*, *tcdB*, *cdtA,* and *cdtB* and housekeeping gene *tpi* of *C. difficile* using the primers shown in Table S2. Only the *tcdB+* gene was considered to be present in the fecal samples (29). Subsequent analyses were performed using *tcdB+* as the criterion for determining the CDI in fecal samples. PCR amplification and the reaction procedure were performed as previously described (8).

### Bacterial culture, coinfection, and antibiotic susceptibility tests

All diarrheal feces were inoculated on selective cycloserine-cefoxitin-fructose agar plates (CCFA, Solarbio) and incubated in an anaerobic jar (Mitsubishi, Japan) at 37°C for 48 h. The suspected colonies were purified by using BHI agar (Oxoid, UK) and identified by MALDI-TOF-MS (Bruker).

In addition to confirming *C. difficile* infection, we screened fecal samples for other enteric pathogens. A wide range of pathogens, including diarrheagenic *E. coli*, *Salmonella*, *Shigella*, *Yersinia*, *rotavirus*, *norovirus*, *astrovirus,* and *adenovirus*, were tested (8, 9).

Metronidazole (MTZ), vancomycin (VA), amoxicillin/clavulanic acid (AMC), erythromycin (E), ciprofloxacin (CIP), tetracycline (TE), clindamycin (DA), ceftazidime (CAZ), amoxicillin (AML), cefotaxime (CTX), imipenem (IPM), and gentamicin (CN) were used for antibiotic susceptibility tests with E-test strips (Liofilchem). The method was based on a previous study (30). The 0.85% saline was used to adjust the 1 McFarland standard of strains and swabbed onto Mueller-Hinton (MH) agar supplemented with 5% horse blood, heme (5 mg/mL), and Vitamin K1 (1 mg/mL). Plates were incubated anaerobically at 37°C, and MIC breakpoints were read after 48 h cultures. The interpretations of the results were determined according to Clinical and Laboratory Standards Institute (CLSI) M11-A7 and M100-S24. ATCC700057 of the *C. difficile* strain was used as a quality control.

### Multilocus sequence typing and genome sequencing

MLST was performed on all isolates using the method developed by Griffiths et al. (31). The experimental procedure was performed as previously described (8). Genomic sequencing was performed on the Illumina MiSeq platform using $2 \times 150$ bp paired-end reads. The libraries were constructed using a Nextera XT DNA Library Prep Kit according to our previous study (30). Bioinformatics analysis of the *C. difficile* genome was performed as previously described (30). The phylogenetic tree of the core SNP alignment was built based on the SNPs of concatenated core genes with Snippy (32).

## Genome collinearity

R20291 (ST1/RT027, GenBank accession: CP029423) and CD21062 (ST11/RT078, GenBank accession: CP033216) were selected as the reference genomes. The YNCD22.413 (ST37 genotype, shown as CD413) strain was used as the research object, and the genome was analyzed for collinearity using Mauve software (33).

## Genomic variation and annotation

R20291 and CD21062 were used as reference genomes. The single-nucleotide polymorphisms (SNPs) and insertions and deletions (InDels) of CD413 were analyzed via GATK software (34). On the basis of the results of the positioning of the clean reads in the reference genome, redundant reads were filtered via SAMtools (V1.9) (35). Variant detection for SNPs and InDels was then performed using the HaplotypeCaller algorithm of GATK. SnpEff software was used for variant annotation and prediction of variant effects (36). Sequence comparison of all variant gene sets with the KEGG database was performed via BLAST software to obtain the annotation information of variant genes (37).

## Mobile genetic element analysis

ISEScan was used to predict insertion sequences in the genomes of representative strains (38); BacAnt was used to predict transposons (39); plasmids were analyzed via Plasmid-Finder (40); phages were analyzed via ProphET and PhiSpy (41); and Mobile Element Finder was used to predict integrative and conjugative elements in genomes (integrative and conjugative elements, ICE) (42).

## RNA-seq of representative strains

R20291, CD21062, YNCD22.279 (ST54, shown as CD279), and YNCD22.413 (ST37, shown as CD413) were selected as representative strains for RNA-seq. Four strains of *C. difficile* were grown on BHI agar, and the McF was adjusted to 0.5 by using PBS. Then, 200 µL of each bacterial suspension was added to 5 mL of TY medium and incubated at 37°C for 24 h. The bacteria were centrifuged at $12,000 \times g$ for 5 min, after which the supernatants were removed. Total RNA was extracted from the bacteria using RNAiso Plus (TaKaRa) according to the manufacturer's instructions. Each strain included three biological replicates. RNA library construction, sequencing methods, and bioinformatics analysis were based on a previous study (43). RNA-seq was verified by RT-qPCR according to the $2^{-\Delta\Delta CT}$ method, and *nisR* and *RS16530* were selected as differentially expressed genes, while *rpoA* was used as an internal reference gene. The primers used in this study are shown in Table S2.

## Biofilm formation assay

Biofilm testing was performed according to previous reports with some modifications (44, 45). In general, 6-well polystyrene plates were used, and each *C. difficile* strain was cultured on BHI agar as mentioned above. Two hundred microliters of each bacterial suspension (0.5 McF) was added to 2 mL of BHI broth in 6-well plates and incubated under anaerobic conditions at 37°C for 24 h. The supernatant was removed after incubation, and the wells were washed twice with PBS and allowed to dry for 10 min. The biofilm was stained with 1 mL of 0.2% crystal violet (CV) at 37°C for 30 min. Then, the CV was removed, and the wells were washed twice with PBS. Bacterial biofilms were observed by light microscopy. An ethanol: acetone (80:20) mixture was used to extract the CV dye, and the intensity of the extracted dye was measured at 570 nm by a spectrophotometer (Bio-Rad).

## Toxin gene expression and cytotoxicity to cells

The expression levels of *tcdA* (TcdA) and *tcdB* (TcdB) in the four *C. difficile* ST strains were determined via RT-qPCR and ELISA, respectively, as described in our previous study (27). The isolates were cultured in TY media at 37°C for 24 h. Total RNA was extracted by RNAiso Plus as described above, and the relative expression of toxin genes was analyzed using a One Step TB Green PrimeScript RT−PCR Kit II (TaKaRa). *rpoA* was used as the internal control. The primers used are listed in Table S2. The supernatants of the cultures in TY medium were filtered through 0.45 µm filters and then coated with 96-well plates at 4°C overnight. After washing with PBS, the plate was blocked with 5% skim milk at 37°C for 2 h. The primary and secondary antibodies were incubated as described previously (27). A TMB chromogenic reagent kit (Sangon) was used to determine the absorbance at 450 nm.

Monolayer Hep-2 cells were prepared by seeding 6-well plates at 37°C in 5% $CO_2$ for 24 h. *C. difficile* strains ($5 \times 10^5$ CFU/mL) were added to Hep-2 cells in 6-well plates and incubated for 3 h anaerobically. The cells were examined by light microscopy (Motic), and three biological replicates were performed in triplicate. The supernatants of the co-cultures were used to measure cytotoxicity by measuring LDH release (Biosharp) in the cells.

## Survival rate of Hep-2 cells

The procedure for the *C. difficile* and cell interaction experiments was the same as that described above. The cell survival rate was estimated by using a TC20 automated cell counter (Bio-Rad) following the manufacturer's instructions. An aliquot of the cell suspension (10 µL) was added to 10 µL of 0.4% trypan blue solution in a test tube and mixed by gently pipetting. Ten microliters were then transferred into the counting chamber of a TC20 counting slide and analyzed via a TC20 cell counter. Three biological replicates were performed per experiment, and the experiment was repeated three times.

## Statistical analysis

Data analysis was performed with IBM SPSS software (version 20.0) and GraphPad Prism 8. Logistic regression analysis was used to identify independent risk factors. Two-sided significance was assessed for all variables, applying a cut-off value of $P < 0.05$.

## ACKNOWLEDGMENTS

We sent our manuscript to American Journal Experts (www.aje.com) for English language revision.

This work was supported by National Natural Science Foundation of China (grant No. 82360398), National Key Research and Development Program of China (grant No. 2023YFC2308800), Yunnan Fundamental Research Project (grant No. 202301AT070160 and 202401AT070054), Yunnan Provincial Technological Innovation Talent Plan (grant No. 202405AD350003), Support Program of Young and Middle-aged Talents in the Field of Infectious Diseases Control, and Prevention of Chinese Preventive Medicine Association (grant No. CPMA2024CRBFK) and Infectious Disease Spectrum and Epidemiology Project of YNCDC (grant No. YNAPM2025-003).

X.F. and Y.W. designed the research; W.G., F.L., L.B., and W.Z. performed the research; W.G., F.L., S.J., J.L., Y.Z., and J.Y. analyzed the data; F.L. and W.G. wrote the paper.

The authors declare that the research was conducted in the absence of any commercial or financial relationships that could be construed as potential conflicts of interest.

## AUTHOR AFFILIATIONS

[1]Institute of Acute Infectious Diseases Control and Prevention, Yunnan Provincial Center for Disease Control and Prevention (Yunnan Academy of Preventive Medicine), Kunming, China

[2]Yunnan Key Laboratory of Cross-Border Infectious Disease Control and Prevention and Novel Drug Development, Kunming, China

[3]Department of Respiratory Medicine, the First People's Hospital of Yunnan Province, Kunming, China

[4]National Institute for Communicable Disease Control and Prevention, Chinese Center for Disease Control and Prevention, Beijing, China

[5]National Key Laboratory of Intelligent Tracking and Forecasting for Infectious Diseases, Beijing, China

## AUTHOR ORCIDs

Yuan Wu  http://orcid.org/0000-0002-9436-5929

## FUNDING

| Funder | Grant(s) | Author(s) |
| --- | --- | --- |
| MOST \| National Natural Science Foundation of China (NSFC) | 82360398 | Wenpeng Gu |
| MOST \| National Key Research and Development Program of China (NKPs) | 2023YFC2308800 | Yuan Wu |
| Yunnan Fundamental Research Project | 202301AT070160 and 202401AT070054 | Wenpeng Gu |
| Yunnan Provincial Technological Innovation Talent Plan | 202405AD350003 | Wenpeng Gu |
| Support Program of Young and Middle-aged Talents in the Field of Infectious Diseases Control and Prevention of Chinese Preventive Medicine Association | CPMA2024CRBFK | Wenpeng Gu |
| Infectious Disease Spectrum and Epidemiology Project of YNCDC | YNAPM2025-003 | Wenpeng Gu |

## AUTHOR CONTRIBUTIONS

Wenpeng Gu, Investigation, Methodology, Writing – original draft | Feng Liao, Investigation, Methodology, Writing – original draft | Lulu Bai, Methodology | Wenzhu Zhang, Methodology | Senquan Jia, Data curation | Junrong Liang, Data curation, Software | Yongming Zhou, Data curation | Jianwen Yin, Data curation | Xiaoqing Fu, Conceptualization, Supervision, writing - review and editing | Yuan Wu, Conceptualization, Supervision

## DATA AVAILABILITY

All the data generated or analyzed during this study were included in this published article. The bacterial genome sequence data have been deposited into the National Center for Biotechnology Information (NCBI) https://www.ncbi.nlm.nih.gov with BioProject accession numbers PRJNA785426 and PRJNA1099924. The RNA-seq data of the four strains were also deposited into the NCBI with the BioProject accession number PRJNA1107959.

## ETHICS APPROVAL

The human sample collection and detection protocols were carried out in accordance with relevant guidelines and regulations. All the experimental procedures were approved by the Ethics Review Committee of the Yunnan Provincial Center for Disease Control and

Prevention (No. 2021-09). All adult subjects provided informed consent, and a parent or guardian of any child participant provided informed consent on their behalf.

## ADDITIONAL FILES

The following material is available online.

### Supplemental Material

**Supplemental material (Spectrum02018-24-s0001.pdf).** Tables S1 and S2.

### Open Peer Review

**PEER REVIEW HISTORY (review-history.pdf).**

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
