## [Reviewer comments · Microbiology Spectrum]

Microbiology Spectrum

Changing patterns and biological features of community-acquired *Clostridioides difficile* infection in Southwest China: 7 years of surveillance data

Wenpeng Gu, Feng Liao, Lulu Bai, Wen Zhang, Senquan Jia, Junrong Liang, Yongming Zhou, Jianwen Yin, Xiaoqing Fu, and Yuan Wu

Corresponding Author(s): Yuan Wu, Chinese Center for Disease Control and Prevention

Review Timeline:

Submission Date:	August 15, 2024
Editorial Decision:	December 23, 2024
Revision Received:	February 8, 2025
Accepted:	April 13, 2025

Editor: Anna Moniuszko-Malinowska

Reviewer(s): Disclosure of reviewer identity is with reference to reviewer comments included in decision letter(s). The following individuals involved in review of your submission have agreed to reveal their identity: Maria Radoslavova Pavlova (Reviewer #1); Jianxin Wang (Reviewer #2)

Transaction Report:

DOI: <https://doi.org/10.1128/spectrum.02018-24>

Re: Spectrum02018-24 (Changing patterns and biological features of community-acquired *Clostridioides difficile* infection in Southwest China: 7 years of surveillance data)

Dear Prof. Yuan Wu:

Thank you for the privilege of reviewing your work. Below you will find my comments, instructions from the Spectrum editorial office, and the reviewer comments.

Revision Guidelines

Sincerely,
Anna Moniuszko-Malinowska
Editor
Microbiology Spectrum

Reviewer #1 (Comments for the Author):

Q1: Line 83: .. Detection of *C. difficile* toxin in stool may not be the causative agent in children with diarrhoea, particularly in young children. Could you explain why you focus on the young population in your study, and whether children are examined for another causative agent? Lastly, what are the rates of co-infections, such as enteric bacteria + CD, viruses +CD, protozoa+CD, etc.?

I request the authors to clarify why the Nucleic acid amplification tests were not applied to detect the so important antigen Glutamate dehydrogenase (GDH), which confirms the presence of CD and only then to follow or in parallel to prove GDH, Toxins A/B.

I believe that the authors should not have included such a large group (65.5%) of children under the age of 5 in their study because of the well-known data in the literature about the low incidence of CDI among young children. This is why the lifetime prevalence was so low, only 5.04%. I advise the authors to write a separate manuscript for an epidemiological study on the prevalence of CD among children under 5 years with analysis and incidence of clinical manifestations.

Reviewer #2 (Comments for the Author):

The manuscript presents a 7 - year surveillance study on community - acquired *Clostridioides difficile* infection in Southwest China, analyzing its epidemiological characteristics, genotype change patterns, and biological features of representative strains. The study is of certain scientific and clinical significance. However, there are several areas that require improvement and refinement, as detailed below.

Q1: The antibiotic resistance heatmap in Figure 2 intuitively shows the resistance of different clade and ST type strains to various antibiotics. However, in the result description, the clinical significance of the resistance pattern has not been deeply discussed. For example, for antibiotics with high resistance rates (such as erythromycin, gentamicin, etc.), the impact of these resistance situations on clinical treatment options has not been mentioned, and whether there are specific factors in the local medical environment that contribute to the increase in resistance (such as antibiotic usage habits, hospital infection control measures, etc.) has not been analyzed.

Q2: In the study of strain genotype changes, although the trend of the ST37 genotype gradually becoming the dominant strain was observed, the underlying molecular biological mechanisms behind this change have not been deeply studied. For example, detailed comparison of the genomic differences between the ST37 strain and other genotype strains, including gene sequence variations, insertions and deletions, horizontal gene transfer, etc., has not been conducted to identify the key genes or genomic regions that may contribute to its enhanced adaptability and transmission advantage.

Q3: It is essential to precisely and comprehensively depict the details of the graphical information within the figure legends. For example, in Fig 1c, it is necessary to elucidate the specific meaning of "number" to ensure clarity and avoid ambiguity.

Q4: The visual quality of the pictures requires enhancement. As an illustration, in Fig 1c, the text is occluded, and the closely resembling color palette impairs the readability and interpretability of the figure.

Q5: What are the specific entities or groups that serve as the comparison objects for the statistical differences in Fig 3f/g, Fig 4e, Fig 5a-b, and c-d respectively? This clarification is crucial for a proper understanding of the data analysis and significance.

Q6: Does Fig 5f singularly and comprehensively portray the infection outcomes of three strains like CD413?

Q7: In Fig 5, why is there no analysis of the survival rate of hep G2 cells treated with different strains?

Q8: Please explain the reasons for the inconsistency of the tcdA/B-TcdAB results of Fig. 5a-5c and Fig. 5b-5d ?

The manuscript presents a 7-year surveillance study on community - acquired *Clostridioides difficile* infection in Southwest China, analyzing its epidemiological characteristics, genotype change patterns, and biological features of representative strains. The study is of certain scientific and clinical significance. However, there are several areas that require improvement and refinement, as detailed below.

Q1: The antibiotic resistance heatmap in Figure 2 intuitively shows the resistance of different clade and ST type strains to various antibiotics. However, in the result description, the clinical significance of the resistance pattern has not been deeply discussed. For example, for antibiotics with high resistance rates (such as erythromycin, gentamicin, etc.), the impact of these resistance situations on clinical treatment options has not been mentioned, and whether there are specific factors in the local medical environment that contribute to the increase in resistance (such as antibiotic usage habits, hospital infection control measures, etc.) has not been analyzed.

Q2: In the study of strain genotype changes, although the trend of the ST37 genotype gradually becoming the dominant strain was observed, the underlying molecular biological mechanisms behind this change have not been deeply studied. For example, detailed comparison of the genomic differences between the ST37 strain and other genotype strains, including gene sequence variations, insertions and deletions, horizontal gene transfer, etc., has not been conducted to identify the key genes or genomic regions that may contribute to its enhanced adaptability and transmission advantage.

Q3: It is essential to precisely and comprehensively depict the details of the graphical information within the figure legends. For example, in Fig 1c, it is necessary to elucidate the specific meaning of "number" to ensure clarity and avoid ambiguity.

Q4: The visual quality of the pictures requires enhancement. As an illustration, in Fig 1c, the text is occluded, and the closely resembling color palette impairs the readability and interpretability of the figure.

Q5: What are the specific entities or groups that serve as the comparison objects for the statistical differences in Fig 3f/g, Fig 4e, Fig 5a-b, and c-d respectively? This clarification is crucial for a proper understanding of the data analysis and significance.

Q6: Does Fig 5f singularly and comprehensively portray the infection outcomes of three strains like CD413?

Q7: In Fig 5, why is there no analysis of the survival rate of hep G2 cells treated with different strains?

Q8: Please explain the reasons for the inconsistency of the *tcdA/B*-TcdAB results of Fig. 5a-5c and Fig. 5b-5d ?

Dear editors:

We have revised our manuscript entitled “Changing patterns and biological features of community-acquired *Clostridioides difficile* infection in Southwest China: 7 years of surveillance data” (Spectrum02018-24) and submitted it to the website. Thanks for the reviewers and editor’s nice comments and suggestions, we have studied the comments carefully and made revision accordingly in the revised manuscript, which we would like to submit for your kind consideration.

Yours sincerely,

Yuan Wu,

Reviewer comments:

Reviewer #1:

Q1: Line 83: .. Detection of *C. difficile* toxin in stool may not be the causative agent in children with diarrhoea, particularly in young children. Could you explain why you focus on the young population in your study, and whether children are examined for another causative agent? Lastly, what are the rates of co-infections, such as enteric bacteria + CD, viruses +CD, protozoa+CD, etc.?

A1: Thanks for the reviewer’s suggestions and advices.

In fact, we performed population-wide surveillance for diarrhea syndrome, and we aimed to study the current status and epidemiological characteristics of *C. difficile* infection in the diarrheal population; therefore, there was no differentiation between children and adult patients in our study. We are also aware that *C. difficile* is considered to be an intestinal colonizing flora in children, especially those under 2 years of age, as suggested by the reviewer. However, our aim was to perform an epidemiological study of *C. difficile* infections in the diarrhea population of the entire region, so we included diarrhea samples from the whole population.

As mentioned by reviewer, we carried out tests on the same fecal samples for viruses such as *rotavirus*, *norovirus*, *astrovirus* and *adenovirus*, as well as bacteria such as diarrheagenic *Escherichia coli*, *Salmonella*, *Shigella*, *Campylobacter* and *Vibrio*, and found that some *C. difficile* infections coinfecting with other pathogens.

We added this part of content in revised manuscript in Materials and Methods section as “In addition to confirming *C. difficile* infection, we screened fecal samples for other enteric pathogens. A wide range of pathogens, including diarrheagenic *E. coli*, *Salmonella*, *Shigella*, *Yersinia*, *rotavirus*, *norovirus*, *astrovirus* and *adenovirus*, were tested (8, 9).”

In results section, we added “The results of *C. difficile* coinfection with other pathogens were shown in Table 3. For all the diarrheal patients, 34 patients were coinfecting with *tcdB+* in stool samples, with more *rotavirus* (29.41%, 10/34) and diarrheagenic *Escherichia coli* (38.24%, 13/34) present. The number of cases of other

pathogens coinfecting with positive *C. difficile* isolates was 10 and was dominated by *rotavirus* (30.00%, 3/10), *adenovirus* (20.00%, 2/10) and diarrheagenic *E. coli* (30.00%, 3/10).

Interestingly, the number of coinfecting patients under 5 years of age accounted for 91.18% (31/34) of the total cases, as shown in Table 3. Among children under 5 years of age, those with CDI coinfections were also predominantly *rotavirus* and diarrheagenic *E. coli*, both in terms of fecal *tcdB*+ and *C. difficile* isolation cultures.”

We added Table 3 of coinfection results in revised manuscript as follow:

Table 3. Coinfection with other enteric pathogens for CDI patients

C. difficile infection	Positive for any virus (case numbers)				Positive for any bacteria (case numbers)		Cases of coinfection
	Rotavirus	Norovirus	Adenovirus	Astrovirus	Diarrheagenic E. coli	Salmonella	
Total cases with diarrhea							
Fecal samples of tcdB +	10	5	4	1	13	1	34
C. difficile isolation+	3	1	2	0	3	1	10
Children under 5 years old							
Fecal samples of tcdB +	10	5	2	1	12	1	31
C. difficile isolation+	3	1	1	0	3	1	9

Q2: I request the authors to clarify why the Nucleic acid amplification tests were not applied to detect the so important antigen Glutamate dehydrogenase (GDH), which confirms the presence of CD and only then to follow or in parallel to prove GDH, Toxins A/B.

A2: Thanks for the reviewer’s suggestions and advices.

We apologize for the confusion caused by our incomplete presentation. As noted by the reviewer, we actually carry out nucleic acid testing of fecal samples for five genes: *tcdA/tcdB/cdtA/cdtB/tpi*. The *tpi* gene, encoding the phosphopropionate isomerase, is used in our laboratory as a housekeeping gene for *C. difficile* detection to demonstrate the presence of *C. difficile* in fecal samples. As the reviewer noted, glutamate dehydrogenase (GDH) has the same role as the housekeeping gene and was also used to detect the presence of *C. difficile* in fecal samples. In our study, we used the positive *tcdB* gene in fecal samples as the final result of CDI, so we did not mention anything related to the *tpi* gene test results in the manuscript. We are very grateful to the reviewer for the suggestions and comments, and we have added the relevant content of *tpi* testing to the revised manuscript.

In Materials and Methods section, we revised as “The genomic DNA of the fecal samples was extracted by using a fecal sample DNA extraction kit (Tiangen, Beijing) according to the manufacturer’s instructions. All the DNA samples were tested for the virulence genes *tcdA*, *tcdB*, *cdtA* and *cdtB* and housekeeping gene *tpi* of *C. difficile*

using the primers shown in Supplementary Table S2. Only the *tcdB+* gene was considered to be present in the fecal samples (29). Subsequent analyses were performed using *tcdB+* as the criterion for determining the CDI in fecal samples. PCR amplification and the reaction procedure were performed as previously described (8).”

In results section, we added related content as “A total of 184 of all the fecal samples were positive for the *tpi* gene, with a positivity rate of 13.45%.”

Q3: I believe that the authors should not have included such a large group (65.5%) of children under the age of 5 in their study because of the well-known data in the literature about the low incidence of CDI among young children. This is why the lifetime prevalence was so low, only 5.04%. I advise the authors to write a separate manuscript for an epidemiological study on the prevalence of CD among children under 5 years with analysis and incidence of clinical manifestations.

A3: Thanks for the reviewer’s suggestions and advices.

Many thanks to the reviewer for the suggestions. As the reviewer noted, *C. difficile* in children under 5 years of age, especially under 2 years, is usually considered to be colonized. This is why the prevalence of CDI in this study was low, and the reviewer’s suggestion is much appreciated. We will write a separate report related to *C. difficile* infection in children under 5 years of age in the next step.

We added the related content in discussion section as follow:

“This study aimed to perform an epidemiological study of *C. difficile* infections in the diarrhea population of Southwest China, so we included diarrhea samples from the whole population. However, such a large group (65.5%) of children patients under the age of 5 in this study, led to the low incidence of CDI and only 5.04% of lifetime prevalence, because of the well-known data in the literature about the CDI among young children (10, 11).”

Reviewer #2: The manuscript presents a 7 - year surveillance study on community - acquired *Clostridioides difficile* infection in Southwest China, analyzing its epidemiological characteristics, genotype change patterns, and biological features of representative strains. The study is of certain scientific and clinical significance. However, there are several areas that require improvement and refinement, as detailed below.

Q1: The antibiotic resistance heatmap in Figure 2 intuitively shows the resistance of different clade and ST type strains to various antibiotics. However, in the result description, the clinical significance of the resistance pattern has not been deeply discussed. For example, for antibiotics with high resistance rates (such as erythromycin, gentamicin, etc.), the impact of these resistance situations on clinical treatment options has not been mentioned, and whether there are specific factors in the local medical environment that contribute to the increase in resistance (such as antibiotic usage habits, hospital infection control measures, etc.) has not been analyzed.

A1: Thanks for the reviewer’s suggestions and advices.

Many thanks to the reviewer for the comments, which have improved and helped us a lot. We added this part of the paragraph to the discussion section as “Moreover, for antibiotics with high resistance rates (such as erythromycin and gentamicin) in Southwest China, the impact of these resistance situations on clinical treatment options should be concerned. Whether there are specific factors in the local medical environment that contribute to the increase in resistance (such as antibiotic usage habits, or hospital infection control measures) need further investigation.”

Q2: In the study of strain genotype changes, although the trend of the ST37 genotype gradually becoming the dominant strain was observed, the underlying molecular biological mechanisms behind this change have not been deeply studied. For example, detailed comparison of the genomic differences between the ST37 strain and other genotype strains, including gene sequence variations, insertions and deletions, horizontal gene transfer, etc., has not been conducted to identify the key genes or genomic regions that may contribute to its enhanced adaptability and transmission advantage.

A2: Thanks for the reviewer’s suggestions and advices.

We added the results of genomic comparisons of the ST37 type with other strains, and the appropriate content in the Methods, Results and Discussion sections were shown as follows:

In Materials and Methods section,

“Genome collinearity

R20291 (ST1/RT027, GenBank accession: CP029423) and CD21062 (ST11/RT078, GenBank accession: CP033216) were selected as the reference genomes. The YNCD22.413 (ST37 genotype, shown as CD413) strain was used as the research object, and the genome was analysed for collinearity using Mauve software (33).

Genomic variation and annotation

R20291 and CD21062 were used as reference genomes. The single nucleotide polymorphisms (SNPs) and insertions and deletions (InDels) of CD413 were analysed via GATK software (34). On the basis of the results of the positioning of the clean reads in the reference genome, redundant reads were filtered via SAMtools (V1.9) (35). Variant detection for SNPs and InDels was then performed using the HaplotypeCaller algorithm of GATK. SnpEff software was used for variant annotation and prediction of variant effects (36). Sequence comparison of all variant gene sets with the KEGG database was performed via BLAST software to obtain the annotation information of variant genes (37).

Mobile genetic element analysis

ISEScan was used to predict insertion sequences in the genomes of representative strains (38); BacAnt was used to predict transposons (39); plasmids were analysed via PlasmidFinder (40); phages were analysed via ProphET and PhiSpy (41); and Mobile Element Finder was used to predict integrative and conjugative elements in genomes (integrative and conjugative elements, ICE) (42).”

In results section,

“Genomic comparison

The results of the genome collinearity analysis revealed that, compared with R20291, CD413 underwent substantial gene rearrangement, as shown in Figure 3a. Gene ectopics and inversions are commonly found in CD413 *C. difficile*, with the majority of genes located in the reverse complementary region compared with the reference region. Gene rearrangements were also very significant in the CD413 strain compared with the CD21062 reference strain, as shown in Figure 3b. Large swaps and inversions of gene positions can also be observed, even to a greater extent than in the R20291 reference genome.

Compared with R20291, a total of 61,589 SNP variants were detected in CD413, of which 48,049 were transitions and 13,540 were transversions, with a Ti/Tv rate of 3.55. Most of the variations were located in the upstream or downstream regions of the gene, accounting for 43.91% and 45.24%, respectively. The proportion of variants occurring in the gene coding region was 8.85%, as shown in Figure 4a. The variant locations of the InDels were also predominantly located upstream of the gene (43.58%) and downstream of the gene (45.54%), whereas the gene coding region accounted for only 3.92%, as shown in Figure 4b. The functional annotations of the SNPs and InDels indicated that the genome functional variants were involved mainly in the metabolism, genetic information and environmental information processing of the strain. Among them, carbohydrate metabolism, amino acid metabolism, and nucleotide metabolism were dominant, as shown in Figure 4c and 4d.

Compared with those in CD21062, a total of 101,900 SNP variants were detected in the CD413 strain, of which 79,180 were transitions and 22,720 were transversions, with a Ti/Tv ratio of 3.48. Most of the variant locations were also located in the upstream or downstream regions of the gene, accounting for 44.34% and 45.57%, respectively. The proportion of mutations in the gene coding region was 8.36%, as shown in Figure 4e. The mutation locations of the InDels were also predominantly located in the upstream (43.64%) and downstream (46.98%) regions of genes, whereas the gene coding region accounted for only 2.21%, as shown in Figure 4f. The results of the functional annotation of the SNPs and InDels indicated that the genome functional variations were involved mainly in the metabolism of the strain, genetic information and environmental information processing. Among them, carbohydrate metabolism, amino acid metabolism, metabolism of cofactors and vitamins and nucleotide metabolism were the main functions, as shown in Figure 4g and 4 h.

We also compared the mobile genetic elements (MGEs) of these representative strains, and the results are shown in Table 5. Among the four representative strains, CD413 had all the MGE elements with insertion sequences, transposons, plasmids, phages and integrative and conjugative elements, most of which were associated with antibiotic resistance and virulence factors of the strain. For example, the transposon Tn925 is associated with tetracycline resistance in the strain; the plasmid DOp1 (repUS43) is associated with tetracycline and β -lactam antibiotic resistance in the strain; and the prophage is associated with extracellular enzymes and adhesion in the strain.”

In discussion section,

“For genomic comparison, the *C. difficile* strain CD413 (ST37 genotype) underwent a high degree of genetic rearrangement. The SNP and InDel regions of the genomic variation were concentrated in the upstream or downstream regions of the coding genes, and these regions were closely related to the gene expression regulation of the bacteria. KEGG functional annotation revealed that these variations were involved mainly in the metabolism of the strain, carbohydrate metabolism, amino acid metabolism, nucleotide metabolism, etc. MGEs analyses revealed that CD413 had a high occurrence of gene transfer events and that these genetic elements conferred antibiotic resistance and some virulence factors to the strain. All these changes in genomic regions may contribute to its enhanced adaptability and transmission advantage to the environment.”

Figure 3, 4 and table 5 were added in the revised manuscript accordingly.

Q3: It is essential to precisely and comprehensively depict the details of the graphical information within the figure legends. For example, in Fig 1c, it is necessary to elucidate the specific meaning of "number" to ensure clarity and avoid ambiguity.

A3: Thanks for the reviewer’s suggestions and advices.

We revised the graphical information of fig 1C, “number” as “Strain numbers” based on the reviewer’s suggestion.

Q4: The visual quality of the pictures requires enhancement. As an illustration, in Fig 1c, the text is occluded, and the closely resembling color palette impairs the readability and interpretability of the figure.

A4: Thanks for the reviewer’s suggestions and advices.

We have changed the color scheme of this figure and optimised the image quality.

Q5: What are the specific entities or groups that serve as the comparison objects for the statistical differences in Fig 3f/g, Fig 4e, Fig 5a-b, and c-d respectively? This clarification is crucial for a proper understanding of the data analysis and significance.

A5: Thanks for the reviewer’s suggestions and advices.

We revised the entities or groups in previous figure 3f/g, figure 4e, figure 5a-b and c-d according to reviewer’s suggestions.

Q6: Does Fig 5f singularly and comprehensively portray the infection outcomes of three strains like CD413?

A6: Thanks for the reviewer’s suggestions and advices.

As the reviewer point out that figure 5f singularly and comprehensively portray the infection outcomes of three strains like CD413.

Q7: In Fig 5, why is there no analysis of the survival rate of hep G2 cells treated with different strains?

A7: Thanks for the reviewer’s suggestions and advices.

We added the survival rate of Hep-2 cells treated with different strains based on the

reviewer's suggestions.

In Materials and Methods section,

“Survival rate of Hep-2 cells

The procedure for the *C. difficile* and cell interaction experiments was the same as that described above. The cell survival rate was estimated by using a TC20 automated cell counter (Bio-Rad) following the manufacturer's instructions. An aliquot of the cell suspension (10 µl) was added to 10 µl of 0.4% trypan blue solution in a test tube and mixed by gently pipetting. Ten microliters were then transferred into the counting chamber of a TC20 counting slide and analysed via a TC20 cell counter. Three biological replicates were performed per experiment, and the experiment was repeated three times.”

In results section,

“The results of the cell viability assay revealed that the average cell survival rates after the interaction of R20291, CD21062, CD279 and CD413 with Hep-2 cells were 17.00%±1.49%, 67.33%±2.34%, 64.89%±2.33% and 63.44%±1.83%, respectively. Hep-2 cells had the lowest cell survival rate after infection with R20291 strain, and the difference was statistically significant from that of the remaining three strains, as shown in Figure 7h. No significant difference was found in the cell survival rate after interaction of the CD21062, CD279 and CD413 strains with cells.”

Figure 5 of previous manuscript was revised as figure 7 accordingly.

Q8: Please explain the reasons for the inconsistency of the *tcdA/B*-TcdAB results of Fig. 5a-5c and Fig. 5b-5d ?

A8: Thanks for the reviewer's suggestions and advices.

Thank you very much for the suggestions and comments. In fact, the TcdA/TcdB protein assay is our own established ELISA. It is possible that low assay sensitivity led to inconsistent results between the relative mRNA quantification and toxin quantification results. However, the trend was consistent between the two methods, both showing that R20291 had the highest virulence production capacity and was significantly different from the other strains. The problem may arise where the CD21062 strain of *tcdA* appears slightly different from its toxin proteins, possibly owing to the sensitivity of the assay and the narrower linear paradigm.

In general, the relative mRNA expression levels of *tcdA/B* and TcdA/B toxins were consistent. From the results, gene transcription and protein expression of both toxins were highest in strain R20291, followed by strain CD21062, while the expression levels of the remaining two strains were similar. Only in the comparison of relative expression of *tcdA/B*, R20291 and CD21062 were statistically different from both CD279 and CD413, whereas in the comparison of TcdA/B proteins, only R20291 was statistically different from CD279 and CD413. However, overall, the mRNA transcription and protein expression trends of the two toxins in the four *C. difficile* strains were consistent. We used ELISA method for the quantification of TcdA/B proteins, which has its own limitations such as detection sensitivity, and there may be some errors in the results when the data are statistically analyzed in comparison with the relative expressions by qPCR method.

We added the content in revised manuscript in discussion section as “From the results, gene transcription and protein expression of both toxins were highest in R20291 strain, followed by CD21062, while the expression levels of the remaining two strains were similar. Although some statistical differences were found between qPCR and ELISA method, the mRNA transcription and protein expression trends of the two toxins in the four *C. difficile* strains were consistent.”

Re: Spectrum02018-24R1 (**Changing patterns and biological features of community-acquired *Clostridioides difficile* infection in Southwest China: 7 years of surveillance data**)

Dear Prof. Yuan Wu:

Your manuscript has been accepted, and I am forwarding it to the ASM production staff for publication. Your paper will first be checked to make sure all elements meet the technical requirements. ASM staff will contact you if anything needs to be revised before copyediting and production can begin. Otherwise, you will be notified when your proofs are ready to be viewed.

Sincerely,
Anna Moniuszko-Malinowska
Editor
Microbiology Spectrum

Reviewer #2 (Comments for the Author):

The author has devised efficacious methods for " Changing patterns and biological features of community-acquired *Clostridioides difficile* infection in Southwest China: 7 years of surveillance data ". The introduction was meticulously crafted, seamlessly integrating preceding research in the field. The methodology section is equally commendable, presenting a lucid and well-organized account of the experimental procedures. The results and discussion sections are articulate and elucidate the findings in a manner that is easily comprehensible to the reader. In conclusion, this scholarly work makes a significant contribution to the existing literature on the subject.